# Association between SNPs in Leptin Pathway Genes and Anthropometric, Biochemical, and Dietary Markers Related to Obesity

**DOI:** 10.3390/genes13060945

**Published:** 2022-05-25

**Authors:** Ricardo Omar Cadena-López, Lourdes Vanessa Hernández-Rodríguez, Adriana Aguilar-Galarza, Willebaldo García-Muñoz, Lorenza Haddad-Talancón, Ma. de Lourdes Anzures-Cortes, Claudia Velázquez-Sánchez, Karla Lucero Flores-Viveros, Miriam Aracely Anaya-Loyola, Teresa García-Gasca, Víctor Manuel Rodríguez-García, Ulisses Moreno-Celis

**Affiliations:** 1Ingeniería en Biotecnología, Facultad de Química, Universidad Autónoma de Querétaro, Querétaro 76010, Mexico; rcadena10@alumnos.uaq.mx; 2Médico Cirujano, Centro de Ciencias de la Salud, Universidad Autónoma de Aguascalientes. Av. Universidad #940, Ciudad Universitaria, Aguascalientes 20100, Mexico; vanessa.hr.uaa@gmail.com; 3Servicio Universitario de Salud, Secretaria de Atención a la Comunidad Universitaria, Universidad Autónoma de Querétaro, Las Campanas, Querétaro 76010, Mexico; adrianaag.ga@gmail.com; 4Laboratorio de Genética Humana, Código 46, S.A. de C.V., Cuernavaca 62498, Mexico; wgarcia@codigo46.com.mx (W.G.-M.); lorenza@codigo46.com.mx (L.H.-T.); lourdes@codigo46.com.mx (M.d.L.A.-C.); cvelazquez@codigo46.com.mx (C.V.-S.); 5Facultad de Enfermería, Universidad Autónoma de Querétaro, Las Campanas, Querétaro 76010, Mexico; karla.flores@uaq.mx; 6Facultad de Ciencias Naturales, Universidad Autónoma de Querétaro, Juriquilla, Querétaro 76230, Mexico; aracely.anaya@uaq.mx (M.A.A.-L.); tggasca@uaq.mx (T.G.-G.); 7Tecnologico de Monterrey, Escuela de Ingeniería y Ciencias, San Pablo, Querétaro 76130, Mexico

**Keywords:** leptin pathway, obesity, single nucleotide polymorphism

## Abstract

Obesity is one of the main public health problems in Mexico and the world and one from which a large number of pathologies derive. Single nucleotide polymorphisms (SNPs) of various genes have been studied and proven to contribute to the development of multiple diseases. SNPs of the leptin pathway have been associated with the control of hunger and energy expenditure as well as with obesity and type 2 diabetes mellitus. Therefore, the present work focused on determining the association between anthropometric markers and biochemical and dietary factors related to obesity and SNPs of leptin pathway genes, such as the leptin gene (LEP), the leptin receptor (LEPR), proopiomelanocortin (POMC), prohormone convertase 1 (PCSK1), and the melanocortin 4 receptor (MC4R). A population of 574 young Mexican adults of both sexes, aged 19 years old on average and without metabolic disorders previously diagnosed, underwent a complete medical and nutritional evaluation, biochemical determination, and DNA extraction from the blood; DNA samples were subsequently genotyped. Association analyses between anthropometric, biochemical, and dietary variables with SNPs were performed using binary logistic regressions (*p*-value = 0.05). Although the sampled population did not have previously diagnosed diseases, the evaluation results showed that 33% were overweight or obese according to BMI and 64% had non-clinically elevated levels of body fat. From the 74 SNP markers analyzed from the five previously mentioned genes, 62 showed polymorphisms within the sampled population, and only 35 of these had significant associations with clinical variables. The risk associations (OR > 1) occurred between clinical markers with elevated values for waist circumference, waist–height index, BMI, body fat percentage, glucose levels, insulin levels, HOMA-IR, triglyceride levels, cholesterol levels, LDL-c, low HDL-c, carbohydrate intake, and protein intake and SNPs of the *LEP*, *LEPR*, *PCSK1*, and *MC4R* genes. On the other hand, the protective associations (OR < 1) were associated with markers including elevated values for insulin, HOMA-IR, cholesterol, c-LDL, energy intake > 2440 Kcal/day, and lipid intake and SNPs of the *LEP* and *LEPR* genes and *POMC*. The present study describes associations between SNPs in leptin pathway genes, revealing positive and negative interactions between reported SNPs and the clinical markers related to obesity in a sampled Mexican population. Hence, our results open the door for the further study of new genetic variants and their influence on obesity.

## 1. Introduction

Obesity is a metabolic disease characterized by a chronic inflammatory process related to the accumulation of ectopic adipose tissue in different areas of the body [1]. Unfortunately, this condition has become more and more frequent in recent years in Mexico, a country that has positioned itself among the nations with the highest rates of obesity in both adults and children [2,3]. According to the 2018 National Survey of Health and Nutrition (ENSANUT) in Mexico, the prevalence of obesity in children aged 5 to 11 years is 20%, compared with 15% among males aged 12 to 19 years. Women between 20 and 29 years old showed an obesity prevalence of 26%, which increased to 46% among women aged 30 to 59 years; when the male population was analyzed, a less pronounced increment was observed, from 24% to 35%. Older adults showed different dynamics; women have a 40% prevalence of obesity, compared to a 26% prevalence among men [4]. Multiple studies reported that obesity is related to several factors, including elevated energy consumption, a sedentary lifestyle, the consumption of alcoholic beverages, smoking, and several genetic factors [5]. For example, studies have described the positive or negative effects of single nucleotide polymorphisms (SNPs) on metabolic pathways and health conditions [6,7].

Leptin (LEP) is a protein with a hormone-like function that is produced and secreted by adipose tissue. Research has demonstrated its importance in controlling the homeostasis of energy regulation; its disruption can cause imbalances in the regulation of food intake, body mass, immune responses, and lipolysis [8,9]. The diversity of leptin’s actions is attributed to its effects on the central nervous system (CNS), which occurs by crossing the blood-brain barrier through a receptor-mediated endocytosis mechanism [10]. Satiety control is regulated by the leptin and melanocortin pathway, in which the LEP secreted by adipocytes interacts with the leptin receptor (LEPR) present in the neurons of the arcuate nucleus of the hypothalamus. Activating this receptor causes the activation of different transcription factors that allow the transcription of proopiomelanocortin (POMC), which is subjected to post-translational modifications, especially to proteolysis regulated by prohormone convertase 1 (PCSK1). The resultant peptides—α and β melanocyte-stimulating hormones—are recognized and activate the melanocortin 4 receptor (MC4R) present in neurons of the paraventricular nucleus, inducing satiety signals and increasing energy utilization (Figure 1) [11,12,13].

Different gene SNPs have been shown to affect leptin and melanocortin pathways, promoting the development of clinical markers of obesity; the polymorphism rs7799039 of the leptin gene has been associated with obesity, and the SNP rs1137101 of the *LEPR* gene has been associated with a risk for the development of type 2 diabetes mellitus in the Turkish population [14]. In the same way, the POMC polymorphism rs934778 has been reported as a risk factor that affects insulin sensitivity [15]. The rs6232 polymorphism of the gene that codes for PCSK1 has also been associated with both childhood and adult obesity [16]; likewise, the rs2229616 polymorphism of the *MC4R* gene has been associated with an increased risk of developing type 2 diabetes mellitus in Saudi patients [17].

Several studies have addressed specific genetic variants of the leptin-melanocortin pathway in the Mexican population, but none of them carried out an integrated evaluation of this pathway. The present work, therefore, focuses on the evaluation of the association of clinical markers related to obesity and SNPs of the *LEP*, *LEPR*, *POMC*, *PCSK1*, and *MC4R* genes in a healthy Mexican population.

## 2. Materials and Methods

### 2.1. Characteristics of the Subjects and Clinical Evaluation

A total sampled population of 574 freshmen from the Universidad Autónoma de Querétaro, both sexes, aged 18 to 30 years without a previous diagnosis of chronic non-communicable diseases, was selected. The participants signed informed consent to perform the clinical evaluation and for the management of the sample and the use of their data for scientific research. The project was approved by the Bioethics Committee of the Facultad de Ciencias Naturales of the Universidad Autónoma de Querétaro (Reference no. 58FCN2020) and performed under the guidelines of the Declaration of Helsinki [18].

All participants underwent an anthropometric assessment that consisted of collecting data from their waist and hip circumference, weight, and height. All anthropometric evaluations were performed by nutritionists, in duplicate, non-consecutively, using previously standardized procedures recommended by the World Health Organization [19].

Body weight and body composition, which are based on the percentages of fat and lean mass, were determined through bioelectrical bioimpedance equipment (SECA mBCA Mod. 514, Hamburg, Germany) previously calibrated with known weight standards. Height was determined with a 2 m stadiometer (SECA-Bodymeter, Mod. 208 Hamburg, Germany) with a separation of 0.1 cm. Height was measured barefoot, ensuring that the heels, calves, buttocks, shoulders, and back of the head were in contact with the wall; measurements were taken according to the “Frankfurt map”. Using the weight and height data, the body mass index (BMI) was calculated. The waist circumference was measured by placing a tape measure (SECA, Mod. 201, Hamburg, Germany) on a line midway between the upper iliac crest and the lower costal margin at the end of a normal expiration. 

For biochemical blood analyses, a fasting blood sample was taken by venipuncture of the arm in 5 mL vacutainer tubes without clotting agents. The blood samples were centrifuged at 2500 rpm for 10 min to obtain the serum needed to perform biochemical analyses of glucose, triglycerides, cholesterol, HDL, and insulin through a colorimetric enzymatic technique (SPINREACT, Girona, Spain) using automated Mindray Mod. BS 120 equipment (Shenzhen, China). Serum LDL-cholesterol concentrations were calculated using the Fridelwald formula (LDL = CT-HDL (TG/5)) in participants with TG < 400 mg/dL.

The presence of obesity risk factors was assessed according to the following anthropometric, biochemical, and clinical indicators: body mass index > 25.0 kg/m^2^; waist circumference (women > 80 cm and men > 90 cm); waist–hip index (women > 0.85 and men > 0.95); waist–height ratio > 0.50; body fat percentage (women > 35% and men > 20%); fasting glucose > 100 mg/dL; insulin (>14 µ/mL for women and >11 U/mL for men); HOMA index (>2.9 for women and >2.3 for men); total cholesterol (>200 mg/dL); low-density lipoproteins cholesterol (LDL-c) (>130 mg/dL); high-density lipoproteins cholesterol (HDL-c) (<50 mg/dL for women and <40 mg/dL for men); triglycerides (>150 mg/dL).

Dietary intake information was obtained using a food frequency questionnaire with 116 items that were previously validated by the “Carlos Alcocer Cuarón” FCN-UAQ Nutrition Clinic, Human Nutrition Laboratory (FCN-UAQ). The total energy intake and macronutrient composition were analyzed using the United States Department of Agriculture Food Composition Databases. The total energy intake was dichotomized according to the median of the population. Carbohydrate, protein, and lipid intakes were expressed in percentages and grams. High intakes were considered at >60%, >30%, and >20%, respectively.

### 2.2. Extraction and Quantification of Genetic Material

Genomic DNA was extracted from whole blood samples (200 μL) using the QIAamp 96 DNA blood kit (QIAGEN, Valencia, CA, USA) according to the manufacturer’s protocol and recommendations. The concentration and 260/280 quality ratio for all isolated DNA samples were determined using the Nanodrop spectrophotometer (Wilmington, DE, USA) and stored at −20 °C until use. Purified DNA was used for genotyping using a concentration of 25 ng of DNA/mL with a purity rate of 1.8–2. The samples were diluted to a final stock concentration of 25 ng/mL using nuclease-free water.

### 2.3. Microarray Assay

The Illumina Custom Array was designed including 74 genetic variants for the analyzed genes (Table 1) among other markers. DNA samples (30–50 ng) were genotyped with the Illumina Infinium HTS Automated protocol and the Beadchip Global Screening Array (GSA-24 v1.0) microarray according to the manufacturer’s instructions in the following steps: the whole genome was isothermally amplified, fragmented, precipitated, and resuspended; later, the resuspended samples were hybridized to the array for the enzymatic base extension and fluorescent staining; and finally, the Illumina iScan System recorded the fluorescent data of the beadchips. The genotype calling was determined using the Illumina GenomeStudio Genotyping software and only the samples with call rates greater than 0.95 were considered for this study. The whole protocol was performed in the Código 46 Genetics Laboratories [20].

### 2.4. Genetic and Statistical Analyses

The genotypes were analyzed using GenAlEx to calculate allelic and genotypic frequencies. Alleles with a lower representation within the population (frequency < 0.05) were purged before testing them for the Hardy-Weinberg Equilibrium (HWE) and the presence of private alleles. Recessive genotypes were tested for their statistical significance (*p*-value = 0.05) and compared to both dominant and heterozygous genotypes. 

To evaluate the anthropometric, biochemical, and dietary variables of the population, two groups were classified according to sex, and the means of each group were compared by Student’s *t*-test (*p* ≤ 0.05). Binary logistic regressions were performed to analyze the associations, where all the previously enlisted variables were compared against the SNPs, adjusted according to age and sex. Associations with values of *p* ≤ 0.05 were considered significant. All statistical analyses were performed using the Statistical Package for the Social Sciences (IBM SPSS Statistics for Macintosh, Version 26.0., Armonk, NY, USA: IBM Corp).

## 3. Results

### 3.1. Description of the Study Population

The general characteristics of the population were divided by sexes, where no statistically significant differences were observed (*p* < 0.05) in age, hip circumference, waist-to-height ratio, BMI, serum insulin levels, HOMA-IR, cholesterol, LDL-c, protein intake (%ID), lipid intake (%ID), or daily carbohydrate intake. However, significant statistical differences were observed (*p* < 0.05) in variables associated with sexual dimorphisms, such as height, weight, waist circumference, waist-hip ratio, and percentage of body fat, as well as in biochemical variables, such as glucose, triglycerides, and HDL-c, and dietary intake in daily energy and grams of protein and lipid intake (Table 2).

Although the sampled subjects had not been previously diagnosed with metabolic disorders, we observed that 31% accumulated fat in the abdominal region (Figure 2A) according to waist circumference, 30% had a gynoid-type body distribution according to waist-hip ratio (Figure 2B), and 39% of the studied population showed cardiovascular risk according to the waist-to-height ratio (Figure 2C). Their nutritional status, determined by BMI, showed that the prevalence of obesity was 8%, 25% were overweight, and 67% were of normal weight or were underweight (Figure 2D). According to the percentage of body fat, only 36% of the population had normal or low fat, while 64% had elevated fat (Figure 2E).

### 3.2. Genetic Frequencies

From the 74 SNPs analyzed in this study, only 11 did not show a polymorphic variation within our sampled population: 5 for LEPR, 1 for POMC, 4 for PCSK1, and 1 for MC4R (Table 3). Consequently, 63 SNPs were selected for further analysis. Of these, 35 markers showed significant statistical associations (binary logistic regression, *p* ≤ 0.05) with obesity markers, from which 23 polymorphisms were statistically associated with the risk of any of the evaluated clinical markers (OR > 1) (Table 4), 6 polymorphisms had protective associations with obesity markers (OR < 1) (Table 5), and 3 polymorphisms showed both risk and protective associations with some obesity markers.

### 3.3. Association between Clinical Markers of Obesity and SNPs of Leptin Pathway Genes

Risk associations with anthropometric markers related to obesity showed that the SNPs rs10244329 (OR = 1.965) and rs11760956 (OR = 1.666) found in the *LEP* gene were associated with a large waist circumference, rs10244329 (OR = 2.055) and rs11760956 (OR = 1.523) were associated with the waist–height index, and rs111573261 (OR = 2.399) and rs78862345 (OR = 2.399) from the *LEPR* gene were also associated with the same anthropometric marker. Elevated BMI was associated with rs11760956 (OR = 1.571), and an elevated percentage of body fat was associated with rs10244329 (OR = 2.415), both SNPs from the *LEP* gene.

Statistically significant risk associations were observed between some analyzed SNP markers and obesity-related biochemical markers. A strong positive association between glucose and rs114280901 (OR = 5.169) was found; elevated insulin levels showed associations with the *LEP* gene rs11760956 (OR = 2.175) as well as with rs12035604 (OR = 2.257), rs1137101 (OR = 1.834), and rs4655723 (OR = 2.353) from the *LEPR* gene. Similarly, the HOMA-IR showed a direct association with rs4655724 (OR = 1.671). Associations between lipid profile biomarkers and SNPs from the *LEPR* gene were also found, specifically between elevated triglyceride values and rs6700896 (OR = 1.903) and rs1805096 (OR = 1.872). The rs17392686 marker from the *PCSK1* gene showed a strong association with high levels of total cholesterol (OR = 7.508). Elevated LDL cholesterol was associated with three SNPs from the *LEPR* gene, rs9436301 (OR = 2.612), rs77451629 (OR = 5.216), and rs1171278 (OR = 2.32). Lastly, high-density cholesterol (HDL-c) showed associations with rs12145690 (OR = 1.844), rs9436301 (OR = 1.557), rs11208648 (OR = 3.102), rs970467 (OR = 1.865), rs10128072 (OR = 1.594), and rs1171278 (OR = 1.673) from the *LEPR* gene and rs34114122 (OR = 2.842) from the *MC4R* gene.

Statistical associations between SNPs and dietary factors were found; rs4731426 (OR = 1.724) from the *LEP* gene showed a risk associated with a high intake of carbohydrates as well as with rs77451629 (OR = 5.031), rs2025803 (OR = 1.744), rs2104564 (OR = 3.318), and rs1751492 (OR = 4.262) of the *LEP* gene. Variants rs2229616 (OR = 6.857) from the *MC4R* gene and rs271923 (OR = 3.962) from the *PCSK1* gene were associated with a high protein intake.

Interestingly, no statistically significant protective associations were observed with anthropometric obesity markers. However, rs11208659 from LEP appeared to be a protective factor for elevated insulin levels (OR = 0.319) and for elevated HOMA-IR (OR = 0.331). The SNPs of the *LEPR* gene, rs1045895 (OR = 0.506) and rs9436748 (OR = 0.462), showed a protective association with elevated total cholesterol, whereas rs10244329 (OR = 0.383) from the *LEP* gene and rs1045895 (OR = 0.287) and rs9436748 (OR = 0.326) were observed to be protective factors against HDL-c. Meanwhile, rs72683113 (OR = 0.549) from the *LEPR* gene showed a protective association with high energy consumption, and rs1137101 (OR = 0.661) from the *LEPR* gene and rs28932472 (OR = 0.658) from the *POMC* gene were seen as protective against high lipid intake (Table 5).

Variants rs10244329 and rs11760956 from the *LEP* gene were directly associated with classic obesity markers (Table 6). These two SNPs were the only markers with statistically significant associations (*p* < 0.05) with body fat percentage, waist circumference, and BMI when the mean values were analyzed using Student’s *t*-test (Appendix A).

## 4. Discussion

The leptin pathway has been consistently associated with food intake and energy expenditure by various authors and associated with obesity and obesity-related diseases [8]. The present study shows multiple associations between genetic variants of the *LEP*, *LEPR*, *POMC*, *PCSK1*, and *MC4R* genes and anthropometric, biochemical, and dietary markers. Waist circumference is an anthropometric marker commonly associated with abdominal obesity and increased morbidity and mortality from associated diseases [21]. The results suggested an association between waist circumference and SNPs from the *LEP* gene, rs10244329 and rs11760956. These two genetic variants have also been associated with obesity markers by other authors, while rs10244329 was associated with body fat index in a study with European adolescents [22], and rs11760956 has been associated with rapid body weight regain [23]. Moreover, these two SNPs were associated with a high waist-to-height ratio along with two more genetic variants from the *LEPR* gene that were not previously analyzed in the literature, rs111573261 and rs78862345. Associations between rs11760956 and elevated BMI, and rs10244329 and elevated body fat percentage were observed and reported in this study.

Concerning biochemical markers, elevated glucose was associated with a genetic variant of the *LEPR* gene, rs114280901, and elevated insulin levels were associated with rs11760956 from the *LEP* gene and with rs1137101 from the *LEPR* gene. Rs1137101 was observed to influence the weight of the mother and the newborn in a Romanian study [24]; likewise, the association between this genetic LEPR variant and a resistance to treatment against breast cancer in a population of overweight Mexican women was also reported [25]. Similarly, our results show an association between elevated insulin levels and rs12035604 and rs4655723 from the *LEPR* gene that was not previously reported. Variants rs6700896 and rs1805096 were found to be associated with triglycerides, whereas rs6700896 was previously associated with increased cardiovascular risk in a meta-analysis reported in 2017 [26] as well as in a published study in a Chinese population [27]. Another study, conducted in the Egyptian population, showed that rs6700896 was associated with non-alcoholic hepatic steatosis and insulin resistance [28], with results suggesting that hypertriglyceridemia is a common marker of non-alcoholic hepatic steatosis and insulin resistance. In a study on a Mexican population diagnosed with morbid obesity, rs1805096 was related to ligament imbalance and was significantly associated with the diagnosed medical condition [29]. Elevated cholesterol had a strong association with a previously unreported variant of the PCSK gene (rs17392686), opening the door for future research on this genetic variant of the gene. 

In our study, elevated LDL-c showed an association with rs9436301, which was previously reported in a study in pregnant women, where it was associated with higher levels of circulating leptin and elevated expression of LEP in the placenta [30]. Moreover, rs1171278 has been associated with the expression of the *LEP* gene and an increment in plasma leptin levels through a genome-wide association study (GWAS) [31]. Our results showed low HDL-c levels were associated with seven LEPR SNPs, one of them not previously reported, as well as another SNP found in the *MC4R* gene. Interestingly, rs12145690 has been observed in a Spanish Mediterranean female population associated with circulating leptin levels, adjusted for BMI [32], and rs970467 has been associated with lipid markers related to kidney cancer [33]. In a systematic review of the Lausanne Cohort 65+, rs10128072 was associated with fat mass and waist circumference, whereas the rs34114122 variant found in the *MC4R* gene was associated with obesity, high fat mass, and high food intake in the Hispanic population [34]. 

Protective associations between SNPs and biochemical markers have been previously reported; variant rs11208659 was identified as a protective factor in a study conducted in Spanish children [35], and in the results of this study, insulin protective factor and observed HOMA-IR values were elevated, which indicates that it may be a genetic marker associated with metabolic alterations and obesity. Our results show rs1045895 to be negatively associated with elevated cholesterol levels; nevertheless, a negative association has been observed with BMI in the American population [36]. No previous studies were found indicating the association of rs9436748 with obesity or any biochemical or anthropometric alteration; however, it has been shown to be a risk factor for breast cancer [37]. Both rs2025803 and rs10749753 have been previously associated as protective factors for elevated LDL-c and have been associated in the same way with low plasma leptin levels [32]. Although published data do not show an association of the SNPs with biochemical markers, these previous studies had a different approach from the one used in our research. Therefore, these associations have certainly not been explored. When analyzing the food intake data, rs4731426 of the *LEP* gene showed a negative association with a high consumption of carbohydrates. In a previous study performed on a population of South India, rs4731426 was associated with obesity and increased body weight gain [38], and its relationship with obesity has already been reported in other populations [23].

## 5. Conclusions

The analyzed population in this study was not previously diagnosed with metabolic disorders, so the associations found between SNPs and clinical variables are highly relevant to understanding future pathologies in the studied population. Although the reported markers associated with energy and macronutrient intake have been previously described, the strong associations found in the present study are unique and provide new insights into the association between clinical and genetic markers, ascertaining the influence of the SNPs of the leptin pathway in the individual imbalances in the evaluated clinical markers.

## Figures and Tables

**Figure 1 genes-13-00945-f001:**
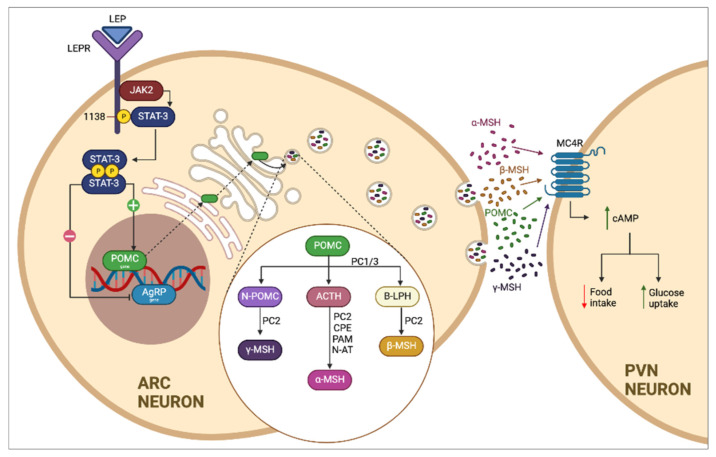
Classical leptin–melanocortin pathway. The melanocortin pathway is regulated by the production of leptin (LEP) and its receptor (LEPR) present in neurons of the arcuate nucleus of the hypothalamus. LEP induces the expression of proopiomelanocortin (POMC) by activating the JAK-STAT pathway, which is transported by cellular cisterns and degraded by specific enzymes present in the cell, including prohormone convertase 1 (PCSK1, PC1/3), which promotes the formation of α and β melanocyte-stimulating hormones (α/β-MSH) and which are recognized by the melanocortin 4 receptor (MC4R) present in neurons of the paraventricular nucleus of the hypothalamus, which induces the sensation of satiety and increased energy use.

**Figure 2 genes-13-00945-f002:**
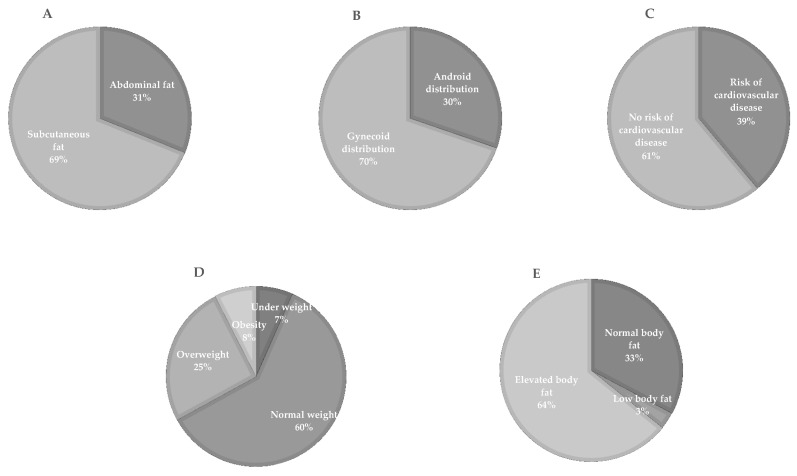
Prevalence of anthropometric markers in the population. (**A**) The corporal mass distribution according to waist circumference is shown. (**B**) shows the distribution of body fat according to the waist-hip ratio. (**C**) shows the percentage of the population at risk of cardiovascular disease according to the waist-to-height ratio. (**D**) The percentage of the population with low weight, normal weight, overweight, and obesity according to BMI is shown. (**E**) shows the prevalence of low, normal, and high body fat.

**Table 1 genes-13-00945-t001:** List of 74 genetic variants evaluated in the study.

Gene	Genetic Variant	Alleles	Functional Consequence
*LEP*	rs4731426	G > C	Intron variant
	rs12706832	A > G	Intron variant
	rs10244329	A > T	Intron variant
	rs11760956	G > A	Intron variant
	rs2071045	T > C	Intron variant
*LEPR*	rs12145690	A > C	5 prime UTR variant, intron variant, upstream transcript variant
	rs12077210	C > T	Intron variant, downstream transcript variant, downstream transcript variant, genic upstream transcript variant
	rs9436301	T > C	upstream transcript variant, intron variant, downstream transcript variant
	rs11208648	A > G	upstream transcript variant, intron variant, downstream transcript variant
	rs1045895	G > A	downstream transcript variant, intron variant, upstream transcript variant, 3 prime UTR variant
	rs75417229	G > A	upstream transcript variant, intron variant
	rs10889551	A > G	Genic upstream transcript variant, intron variant
	rs970467	C > T	Intron variant, genic upstream transcript variant
	rs9436746	A > C	Genic upstream transcript variant, intron variant
	rs77451629	G > A	Genic upstream transcript variant, intron variant
	rs9436748	G > T	Genic upstream transcript variant, intron variant
	rs114280901	G > A	Intron variant, genic upstream transcript variant
	rs138473950	C > T	Intron variant, genic upstream transcript variant
	rs17412347	C > T	Intron variant, genic upstream transcript variant
	rs17127656	C > T	Intron variant, genic upstream transcript variant
	rs2025804	G > A	Intron variant, genic upstream transcript variant
	rs2025803	A > G	Intron variant, genic upstream transcript variant
	rs17127673	A > G	Intron variant, genic upstream transcript variant
	rs10128072	A > C	Genic upstream transcript variant, intron variant
	rs78650744	T > C	Genic upstream transcript variant, intron variant
	rs74082072	T > C	Genic upstream transcript variant, intron variant
	rs2767485	T > C	Intron variant, genic upstream transcript variant
	rs11208659	T > C	Genic upstream transcript variant, intron variant
	rs1171278	C > T	Intron variant, genic upstream transcript variant
	rs1751492	C > T	Intron variant
	rs1782754	G > A	Intron variant
	rs10749753	A > G	Intron variant
	rs1177681	G > A	Intron variant
	rs111573261	A > G	Intron variant
	rs117291834	A > G	Intron variant
	rs74986928	T > G	Intron variant
	rs78862345	G > A	Intron variant
	rs12059300	G > A	Intron variant
	rs7413467	A > G	Intron variant
	rs150025527	A > G	Intron variant
	rs77715828	A > G	Intron variant
	rs12038998	C > A	Intron variant
	rs10789188	A > G	Intron variant
	rs61781283	G > A	Intron variant
	rs72683113	T > C	Intron variant
	rs12035604	T > C	Intron variant
	rs1137101	A > G	Missense variant, coding sequence variant
	rs6700201	C > T	Intron variant
	rs1805134	T > C	Synonymous variant, coding sequence variant
	rs17097193	T > C	Intron variant
	rs79843967	A > G	Intron variant
	rs4606347	G > A	Intron variant
	rs1805094	G > C	Missense variant, coding sequence variant
	rs4655723	C > T	Intron variant
	rs10889569	A > T	Intron variant
	rs4567312	C > T	Intron variant
	rs6700896	C > T	Intron variant
	rs1805096	G > A	Genic downstream transcript variant, synonymous variant, coding sequence variant
	rs1892534	C > T	Genic downstream transcript variant, 3 prime UTR variant
*POMC*	rs28932472	G > C	Coding sequence variant, missense variant
	rs7591899	G > A	Intron variant
	rs934778	A > G	Intron variant
*PCSK1*	rs144800629	G > A	3 prime UTR variant, intron variant
	rs13169290	G > A	Intron variant
	rs271923	T > C	Intron variant
	rs156016	A > G	Intron variant
	rs1498928	A > G	Intron variant
	rs183045011	A > G	Missense variant, intron variant, coding sequence variant
	rs6232	T > C	Intron variant, coding sequence variant, missense variant
	rs17392686	A > G	Intron variant
	rs140520429	G > A	Coding sequence variant, intron variant, missense variant
	rs6889272	C > T	Intron variant
*MC4R*	rs2229616	C > T	Coding sequence variant, missense variant
	rs34114122	T > G	5 prime UTR variant

**Table 2 genes-13-00945-t002:** Anthropometric, biochemical, and dietary variables of the sampled population.

	Sex	Mean	Standard Deviation	*p*-Value
Age (years)	Women	19.07	1.78	0.17
	Men	19.29	2.08	
Height (cm)	Women	159.71	6.32	0.00
	Men	171.36	7.03	
Weight (Kg)	Women	60.03	12.51	0.00
	Men	70.63	13.29	
Waist circumference (cm)	Women	78.38	12.06	0.00
	Men	83.90	11.45	
Hip circumference (cm)	Women	96.49	8.56	0.51
	Men	96.95	8.07	
Waist–Hip ratio	Women	0.81	0.07	0.00
	Men	0.86	0.06	
Waist–Height index	Women	0.49	0.08	0.95
	Men	0.49	0.07	
BMI (Kg/m^2^)	Women	23.56	4.70	0.19
	Men	24.05	4.17	
Body Fat (%)	Women	31.48	7.42	0.00
	Men	21.10	8.11	
Glucose (mg/dL)	Women	82.11	8.86	0.00
	Men	84.97	8.84	
Insulin (ug/mL)	Women	8.14	6.19	0.42
	Men	7.74	5.16	
HOMA-IR	Women	1.62	1.21	0.88
	Men	1.63	1.16	
Triglycerides (mg/dL)	Women	96.95	53.72	0.00
	Men	114.44	73.15	
Cholesterol (mg/dL)	Women	157.25	27.31	0.96
	Men	157.38	32.59	
LDL-c (mg/dL)	Women	84.38	22.14	0.31
	Men	86.35	24.80	
HDL-c (mg/dL)	Women	53.22	13.37	0.00
	Men	48.11	11.10	
Energy intake (Kcal/day)	Women	2407.62	913.14	0.01
	Men	2597.30	887.05	
Protein intake (g)	Women	98.08	38.13	0.00
	Men	107.97	45.31	
Protein intake (% ID)	Women	16.53	2.93	0.87
	Men	16.57	3.12	
Lipid intake (g)	Women	79.67	32.97	0.00
	Men	88.01	36.96	
Lipid intake (% ID)	Women	29.87	5.17	0.39
	Men	30.26	5.98	
Carbohydrate intake (g)	Women	332.55	138.75	0.13
	Men	349.37	126.35	
Carbohydrate intake (% ID)	Women	55.06	7.61	0.15
	Men	54.12	8.12	

Student’s *t*-test of statistical significance, *p*-value < 0.05.

**Table 3 genes-13-00945-t003:** Genetic frequencies of evaluated variants.

Gene	SNP	Genotype	Frequency (%)	Gene	SNP	Genotype	Frequency (%)	Gene	SNP	Genotype	Frequency (%)
*LEPR*	rs12145690	AA	23.7	*LEPR*	rs74986928	TG	1.7	*POMC*	rs7591899	GG	100.0
		AC	51.9			TT	98.3			AG	0.0
		CC	24.4			GG	0.0			AA	0.0
	rs12077210	CC	96.9		rs78862345	AA	1.4		rs934778a	AA	56.8
		TC	3.1			AG	4.2			AG	34.7
		TT	0.0			GG	94.4			GG	8.5
	rs9436301	CC	9.4		rs12059300	AA	1.4	*PCSK1*	rs144800629	GG	100.0
		TC	41.8			AG	12.2			AG	0.0
		TT	48.8			GG	86.4			AA	0.0
	rs11208648	AA	95.5		rs7413467	AA	42.5		rs13169290	AA	3.1
		AG	4.5			AG	43.6			AG	18.5
		GG	0.0			GG	13.9			GG	78.4
	rs1045895	AA	8.4		rs150025527	AA	99.8		rs271923	CC	45.3
		AG	42.0			AG	0.2			TC	44.1
		GG	49.7			GG	0.0			TT	10.5
	rs75417229	AG	0.3		rs77715828	AA	99.8		rs156016	AA	30.7
		GG	99.7			AG	0.2			AG	47.0
		AA	0.0			GG	0.0			GG	22.2
	rs10889551	AA	13.8		rs12038998	AA	17.2		rs1498928	AA	3.1
		AG	43.6			AC	34.5			AG	29.4
		GG	42.7			CC	48.3			GG	67.6
	rs970467	CC	67.6		rs10789188	AA	10.8		rs183045011	AA	100.0
		CT	21.3			AG	42.5			AG	0.0
		TT	11.1			GG	46.7			GG	0.0
	rs9436746	AA	18.1		rs61781283	AA	5.1		rs6232	TC	3.1
		AC	42.9			AG	35.2			TT	96.9
		CC	39.0			GG	59.8			CC	0.0
	rs77451629	AG	1.9		rs72683113	CC	0.3		rs17392686	AA	99.2
		GG	98.1			TC	10.6			AG	0.8
		AA	0.0			TT	89.0			GG	0.0
	rs9436748	GG	47.4		rs12035604	CC	15.3		rs140520429	AG	0.3
		GT	43.4			TC	42.7			GG	99.7
		TT	9.2			TT	42.0			AA	0.0
	rs114280901	AG	3.3		rs1137101	AA	28.9		rs6889272	CC	1.0
		GG	96.7			AG	46.3			CT	7.0
		AA	0.0			GG	24.7			TT	92.0
	rs138473950	CC	98.3		rs6700201	CC	77.0	*MC4R*	rs2229616	CC	98.8
		TC	1.7			CT	19.7			TC	1.2
		TT	0.0			TT	3.3			TT	0.0
	rs17412347	CC	98.3		rs1805134	CC	5.1		rs34114122	TT	97.3
		TC	1.7			CT	18.6			TG	2.7
		TT	0.0			TT	76.3			GG	0.0
	rs17127656	CC	92.2		rs17097193	CC	0.7				
		TC	7.8			TC	3.7				
		TT	0.0			TT	95.6				
	rs2025804	AA	47.0		rs79843967	AA	97.6				
		AG	38.3			AG	2.3				
		GG	14.6			GG	0.2				
	rs2025803	AA	67.8		rs4606347	AA	1.4				
		AG	19.7			AG	29.4				
		GG	12.5			GG	69.2				
	rs17127673	AA	55.2		rs1805094	CC	1.4				
		AG	39.2			CG	29.4				
		GG	5.6			GG	69.2				
	rs10128072	AA	55.2		rs4655723	CC	46.0				
		AC	39.2			TC	40.8				
		CC	5.6			TT	13.2				
	rs78650744	TC	2.4		rs10889569	AA	16.2				
		TT	97.6			AT	44.3				
		CC	0.0			TT	39.5				
	rs74082072	CC	0.2		rs4567312	CC	96.2				
		TC	12.0			TC	3.7				
		TT	87.8			TT	0.2				
	rs2767485	CC	9.1		rs6700896	CC	23.9				
		CT	26.1			TC	47.9				
		TT	64.8			TT	28.2				
	rs11208659	CC	0.3		rs1805096	AA	27.5				
		TC	13.4			AG	48.8				
		TT	86.2			GG	23.7				
	rs1171278	CC	50.9		rs1892534	CC	23.2				
		TC	40.8			TC	49.0				
		TT	8.4			TT	27.9				
	rs1751492	CC	11.3	*LEP*	rs4731426	CC	17.7				
		TC	46.2			GC	56.5				
		TT	42.5			GG	25.6				
	rs1782754	AA	46.7		rs12706832	AA	52.0				
		AG	42.9			AG	26.8				
		GG	10.5			GG	21.1				
	rs10749753	AA	56.8		rs10244329	AA	13.6				
		AG	33.1			TA	28.2				
		GG	10.1			TT	58.1				
	rs1177681	AA	47.7		rs11760956	AA	19.5				
		AG	42.0			AG	45.5				
		GG	10.3			GG	34.8				
	rs111573261	AA	94.4		rs2071045	CC	4.2				
		AG	5.6			TC	32.6				
		GG	0.0			TT	63.0				
	rs117291834	AA	97.2	*POMC*	rs28932472	CC	34.7				
		AG	2.8			CG	23.5				
		GG	0.0			GG	41.8				

**Table 4 genes-13-00945-t004:** Significant statistical associations between clinical markers of obesity and SNPs of LEP, LEPR, POMC, PCSK1, and MC4R, considered to be risk factors.

Clinical Marker	Gene	SNP	OR	CI95%		*p*-Value
Large waist circumference	LEP	rs10244329	1.965	1.08	3.577	0.027
	LEP	rs11760956	1.666	1.124	2.471	0.011
Large waist-height index	LEP	rs10244329	2.055	1.187	3.557	0.01
	LEP	rs11760956	1.523	1.059	2.19	0.023
	LEPR	rs111573261	2.399	1.148	5.015	0.02
	LEPR	rs78862345	2.399	1.148	5.015	0.02
Elevated BMI	LEP	rs11760956	1.571	1.059	2.332	0.025
Elevated body fat (%)	LEP	rs10244329	2.415	1.395	4.181	0.002
High glucose	LEPR	rs114280901	5.169	1.061	25.18	0.042
Elevated insulin levels	LEP	rs11760956	2.175	1.187	3.985	0.012
	LEPR	rs12035604	2.257	1.3	3.916	0.004
	LEPR	rs1137101	1.834	0.998	3.369	0.051
	LEPR	rs4655723	2.353	1.374	4.03	0.002
High HOMA-IR	LEPR	rs4655724	1.671	0.991	2.816	0.054
Elevated triglyceride levels	LEPR	rs6700896	1.903	1.027	3.523	0.041
	LEPR	rs1805096	1.872	1.011	3.467	0.046
Elevated cholesterol levels	PCSK1	rs17392686	7.508	1.196	47.127	0.031
Elevated LDL-c	LEPR	rs9436301	2.612	1.07	6.377	0.035
	LEPR	rs77451629	5.216	1.047	25.969	0.044
	LEPR	rs1171278	2.32	0.981	5.485	0.055
Low HDL-c	LEPR	rs12145690	1.844	1.182	2.876	0.007
	LEPR	rs9436301	1.557	1.09	2.224	0.015
	LEPR	rs11208648	3.102	1.349	7.13	0.008
	LEPR	rs970467	1.865	1.284	2.71	0.001
	LEPR	rs10128072	1.594	1.115	2.278	0.011
	LEPR	rs1171278	1.673	1.17	2.391	0.005
	MC4R	rs34114122	2.842	0.985	8.197	0.053
High carbohydrate intake	LEP	rs4731426	1.724	1.065	2.791	0.027
High protein intake	LEPR	rs77451629	5.031	1.426	17.758	0.012
	LEPR	rs2025803	1.744	1.019	2.986	0.042
	LEPR	rs2104564	3.318	1.011	10.883	0.048
	LEPR	rs1751492	4.262	1.016	17.88	0.048
	MC4R	rs2229616	6.857	1.485	31.66	0.014
	PCSK1	rs271923	3.962	0.943	16.645	0.05

**Table 5 genes-13-00945-t005:** Significant statistical associations between clinical markers of obesity and SNPs of LEP, LEPR, POMC, PCSK1, and MC4R: protective factors.

Clinical Marker	GENE	SNP	OR	CI95%		*p*-Value
Elevated insulin levels	LEPR	rs11208659	0.319	0.112	0.906	0.032
High HOMA-IR	LEPR	rs11208659	0.331	0.117	0.94	0.038
Elevated cholesterol levels	LEPR	rs1045895	0.506	0.263	0.972	0.041
	LEPR	rs9436748	0.462	0.24	0.887	0.02
Elevated LDL-c	LEP	rs10244329	0.383	0.154	0.955	0.039
	LEPR	rs1045895	0.287	0.112	0.731	0.009
	LEPR	rs9436748	0.326	0.134	0.796	0.014
	LEPR	rs2025803	0.274	0.081	0.929	0.038
	LEPR	rs10749753	0.393	0.154	1.003	0.051
Energy intake > 2440 Kcal/day	LEPR	rs72683113	0.549	0.312	0.963	0.037
High lipid intake	LEPR	rs1137101	0.661	0.452	0.966	0.032
	POMC	rs28932472	0.658	0.466	0.928	0.017

**Table 6 genes-13-00945-t006:** Determination of the influence of SNPs on obesity.

SNP	Obesity marker	Model	Mean	S.D.	*p*-Value
LEP					
rs10244329	Body fat (%)	XX	24.39	10.51	0.045
		Xx + xx	26.73	9.07	
	Waist circumference	XX	77.80	13.29	0.011
		Xx + xx	81.52	11.81	
	BMI	XX	22.60	5.09	0.014
		Xx + xx	23.95	4.29	
rs11760956	Body fat (%)	XX	25.05	9.52	0.011
		Xx + xx	27.15	9.11	
	Waist circumference	XX	79.23	11.77	0.009
		Xx + xx	81.97	12.14	
	BMI	XX	23.03	4.45	0.004
		Xx + xx	24.15	4.36	

## Data Availability

Not applicable.

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
