# Peer review of "Association between SNPs in Leptin Pathway Genes and Anthropometric, Biochemical, and Dietary Markers Related to Obesity"

_genes, 2022, doi:10.3390/genes13060945_

Round 1

Reviewer 1 Report

In this manuscript titled, " ssociation between leptin pathway gene SNPs and anthropometric, biochemical, and dietary markers related to obesity.", Ricardo Omar Cadena-López et al., authors focused on determining the association between anthropometric markers, biochemical and dietary factors related to obesity, and SNPs of leptin pathway genes such as leptin gene (LEP), leptin receptor (LEPR), proopiomelanocortin (POMC), prohormone convertase 1 (PCSK1) and the melanocortin 4 receptor (MC4R). This manuscript is written clearly, however, the manuscript appears preliminary.

  1. In this study, authors selected four leptin pathway genes, LEP, LEPR, POMC and MC4R. what is the relationship among these genes?
  2. Authors found many genetic variants, which variants directly related to obesity? Do you do assay to confirm that variant?
  3. What are the downstream targets of LEP, LEPR, POMC and MC4R? Do the variants affect the expression of downstream targets?

Author Response

Dear reviewer, we appreciate your comment. In response to your observations and comments:

Q1. In this study, authors selected four leptin pathway genes, LEP, LEPR, POMC and MC4R. What is the relationship among these genes?

Answer 1:

The relationship among these genes is described in lines 69-83, where a brief description of the pathway triggered by the five evaluated genes is provided. For the leptin-melanocortin pathway to be functionally carried out in the hypothalamus, it is necessary for the participating proteins to act in an appropriate and coordinated manner, therefore, it is necessary to analyze the presence of SNPs in the LEP, LEPR, POMC, PCSK1 and MC4R genes, which codify for key proteins in the described metabolic pathway; and for which their mutations have been associated with monogenic obesity. We have added Figure 1, for the better understanding of the pathway activity and interaction (line 98-105).

Q2. Authors found many genetic variants, which variants directly related to obesity? Do you do assay to confirm that variant?

Answer 2:

We appreciate your comment.

In response to your first question, obesity is a multicausal and multifactorial disease; although currently the BMI, body fat percentage and central adiposity, determined by waist circumference, are direct indicators for the diagnosis of obesity. It should be noted that the aforementioned markers only indicate the body volume, quantity and distribution of body fat and not the metabolic status of the patients. The SNPs specifically associated with these markers are shown in Table 4 of the text, and are only rs10244329 and rs11760956; likewise, to answer the second question, a Student's t-test (p<0.05) was performed, where a difference can be observed between the variants means and the presence of SNPs according to the used model.

To complement, a new analysis was performed for the classic indicators of obesity, for which lines 283-286 and Table 6 (line 287-288) were added to the results section; in the same way, a supplementary table (Table S1) is added with the Student's t-tests, of all the SNPs that were significant in the binary logistic regression, compared with percentage of body fat, waist circumference and BMI, as indicators of obesity.

Q3. What are the downstream targets of LEP, LEPR, POMC and MC4R? Do the variants affect the expression of downstream targets?

Answer 3:

Dear reviewer, we appreciate your comment.

Pathway targets were added in Figure 1 (line 98-105).

I inform you that a revision of the style of the language has been carried out.

Reviewer 2 Report

Dear Authors! It was a great pleasure to read the results of such an extensive study. The "Materials and Methods" section describes in detail all the applied research methods, which are fully consistent with the results obtained. However, there are some technical shortcomings in the text that do not affect the overall positive assessment of the work:

  1. line 117: It is not entirely clear what the ", and height." at the beginning of the paragraph refers to;
  2. there are two paragraphs 2.2 in the text (line 157 and line 168). I suppose 2.2 should be replaced with 2.3 (line 168) and 2.3 should be replaced with 2.4 (line 181).
  3. similar technical error: there are two paragraphs 3.1 in the text (line 199 and line 225). I suppose 3.1 should be replaced with 3.2 (line 225) and 3.2 should be replaced with 3.3 (line 236)

Author Response

Answer

Dear reviewer, we appreciate your comment.

  • The observation of point 1 was corrected.
  • The observations of numbering in the point 2, were fixed.
  • The third observation was corrected.

I inform you that a revision of the style of the language has been carried out.

Round 2

Reviewer 1 Report

Accept in present form